# Phosphorylation of Rec8 cohesin complexes regulates mono-orientation of kinetochores in meiosis I

Yu Liu[1,2], Yu Min[1,2], Yongxin Liu[1,2] , Yoshinori Watanabe[2]

In meiosis I, unlike in mitosis, sister kinetochores are captured by microtubules emanating from the same spindle pole (mono-orientation) and centromeric cohesion mediated by cohesin is protected in the following anaphase I. The conserved meiosis-specific kinetochore protein meikin (Moa1 in fission yeast) associates with polo-like kinase: Plo1 and regulates both mono-orientation and cohesion protection. Although the phosphorylation of Rec8-S450 by Plo1 associated with Moa1 plays a key role in cohesion protection, how Moa1-Plo1 regulates mono-orientation remains elusive. Here, we identify Plo1 phosphorylation sites in the cohesin subunits, Rec8 and Psm3. The non-phosphorylatable mutations at these sites showed specific defects in mono-orientation. These results enabled the genetic dissection of meikin functions at the centromeres.

## Introduction

Sister chromatid cohesion of chromosomes is carried out by a multi-subunit complex, called cohesin, which in fission yeast comprises two SMC (structural maintenance of chromosome) family proteins, Psm1 and Psm3, a kleisin Rad21 subunit and an accessory subunit Psc3 (Tomonaga et al, 2000; Nasmyth, 2001). In proliferating cells, the cohesin complexes are loaded on chromatin largely depending on the loader complex, Mis4-Ssl3 (Murayama & Uhlmann, 2014), which is a potent activator of cohesin's ATPase. Cohesion is established through exchanging the loader complexes with a key cohesin accessory factor, Pds5, which recruits Eso1, a cohesin acetylase (Vaur et al, 2012; Goto et al, 2017). Eso1 acetylates Psm3 at two conserved lysin residues (K105 and K106) to prevent the function of the cohesin releasing factor Wpl1 and thereby establishes cohesion (Feytout et al, 2011; Kagami et al, 2011). Because Mis4 and Pds5 share the binding surface of the kleisin subunit, their binding is mutually exclusive (Kikuchi et al, 2016; Petela et al, 2018). Thus, cohesin loading and cohesion establishment is regulated by distinct cohesin accessory factors, Mis4 and Pds5 (Fig 1A).

Sister chromatid cohesion is established in S phase and maintained until metaphase when the sister chromatids are captured by spindle microtubules from opposite poles. To initiate anaphase, the anaphase-promoting complex (APC/C) triggers the degradation of securin, an inhibitory chaperone for separase that cleaves Rad21 along the entire chromosome in yeast. This triggers the separation of sister chromatids and their movement to opposite poles, a process called equational division (Tomonaga et al, 2000; Nasmyth, 2001). During meiosis, however, the kleisin subunit Rad21 is replaced along the entire chromosomes by a meiotic version Rec8 and one round of DNA replication is followed by two rounds of nuclear division, which results in four haploid nuclei, or gametes (Watanabe & Nurse, 1999; Watanabe et al, 2001). In the first division of meiosis (meiosis I), homologous chromosomes connected by chiasmata are captured by microtubules from opposite poles, whereas sisters are captured from the same pole (mono-orientation) (Hauf & Watanabe, 2004). At the onset of anaphase I, Rec8 cohesin is cleaved by separase along the arm regions, but protected at centromeres throughout meiosis I (cohesion protection) (Buonomo et al, 2000; Kitajima et al, 2003a). Mono-orientation is established by cohesion of the core centromere, whereas cohesion protection acts at the pericentromere (Kitajima et al, 2003b). Cohesion protection is mediated by the centromeric protein shugoshin (Sgo1) (Kitajima et al, 2004), and the meiotic kinetochore protein meikin (Moa1) controls both mono-orientation and cohesion protection (Yokobayashi & Watanabe, 2005; Miyazaki et al, 2017). We previously showed that Moa1-associated Plo1 phosphorylates Rec8-S450 and this phosphorylation is essential for cohesion protection by Sgo1 at the pericentromere (Ma et al, 2021). However, how mono-orientation of sister kinetochores is regulated by Moa1-Plo1 remains largely elusive.

## Results and Discussion

### Cohesin-Mis4 but not cohesin-Pds5 is excessively loaded at the core centromere in *moa1Δ* cells

In meiosis, Rec8 cohesin complexes are loaded at the core centromere to establish cohesion and mono-orientation of

---

[1]School of Bioengineering, Jiangnan University, Wuxi, China   [2]Science Center for Future Foods, Jiangnan University, Wuxi, China

Correspondence: ywatanabe@jiangnan.edu.cn

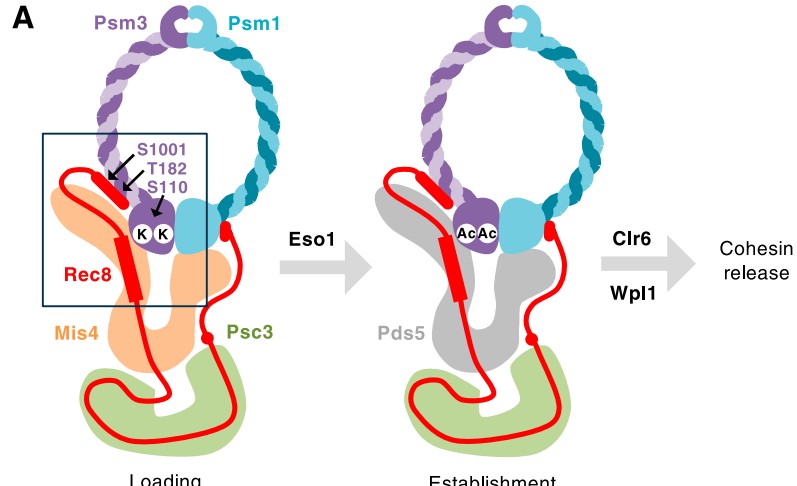

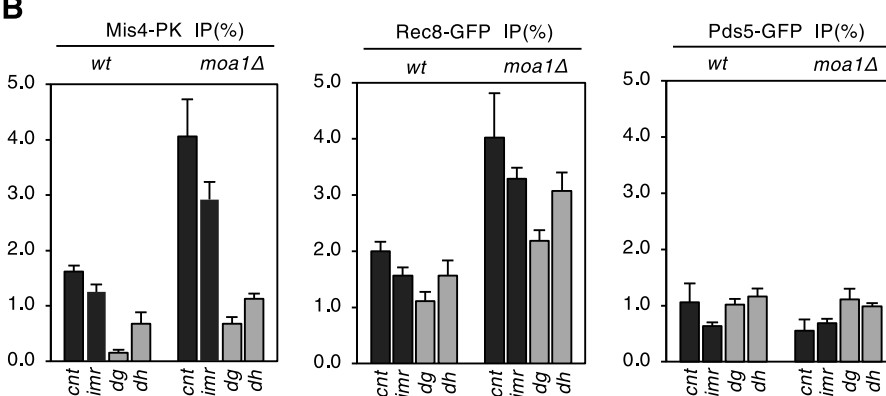

**Figure 1. Cohesin-Mis4 but not cohesin-Pds5 is excessively loaded at the core centromere in *moa1Δ* cells.**

**(A)** Schematic depiction of dynamic regulation of the cohesin complexes. Eso1 acetylates (Ac) two conserved lysin residues (K) of Psm3, which may promote the change of cohesin-Mis4 to cohesin-Pds5. Deacetylation of Psm3 by Clr6 activates cohesin releasing factor Wpl1 and thereby cohesin dissociates from chromosomes, leading to cohesion loss. **(B)** Diploid WT and *moa1Δ* cells were arrested at meiotic prophase I and analyzed for centromeric enrichment profile for Mis4, Rec8, and Pds5 by ChIP assay, using primers that amplify the core centromeric (*cnt* and *imr*) and pericentromeric (*dg* and *dh*) regions. The centromeric enrichment was normalized by the arm enrichment in each sample (see the Materials and Methods section). Error bars represent SD (n = 3 PCR amplifications). Source data are available for this figure.

kinetochores. In *moa1Δ* cells, Rec8 cohesin localization increases at the core centromere although sister chromatid cohesion is abolished at this site (Yokobayashi & Watanabe, 2005; Sakuno et al, 2009). To explore the regulatory mechanisms of Rec8 cohesin at the core centromere, we examined the localization of Rec8, Mis4, and Pds5 by ChIP assay. The data revealed that in *moa1Δ* cells not only the localization of Rec8 but also that of Mis4 increases at the core centromere whereas Pds5 does not (Fig 1B). Based on these ChIP results, we hypothesized that the Mis4-dependent cohesin loading at the core centromere is active in *moa1Δ* cells, but the process of converting cohesin from the Mis4-bound into Pds5-bound form might be hampered. This might be one reason why cohesion is not established or maintained at the core centromere.

### Identification of Plo1 phosphorylation sites in Rec8 required for mono-orientation

Recent cryo-EM studies have revealed how the cohesin complex binds to its loader and DNA (Collier et al, 2020; Higashi et al, 2020; Shi et al, 2020). The studies suggest that the major chromatin binding pocket of the cohesin complexes is formed by the heterodimer of Psm1-Psm3 head domains, Mis4, and the N-terminal helical domain of Rec8 that binds to the coiled-coil region of Psm3 (Fig 1A, square). We explored the possibility that the putative Mis4/

Pds5-binding surface of Rec8 (111–225 a.a.) might be the target of regulation by Moa1-Plo1, which may facilitate the exchange from Mis4 into Pds5. We identified 11 polo-kinase consensus N/Q/E/D-X-S/T sites and four non-consensus S/T in this domain, which were phosphorylated by Plo1 in vitro and some of them were detected also in vivo (Fig S1A–C). To examine the significance of Rec8 phosphorylation by Plo1, we replaced endogenous *rec8⁺* with the non-phosphorylatable mutant *rec8-15A*, in which all 15 serines/threonines in the putative Mis4/Pds5-binding surface of Rec8 were replaced with alanines (Fig 2A). We then examined meiotic chromosome segregation of the mutant. We monitored GFP fluorescence associated with a *lacO* array integrated at the centromere of one homolog of chromosome I (*imr1*::GFP). In *moa1Δ* cells, a minority population (11%) of cells underwent equational segregation at meiosis I (because of defects in mono-orientation), whereas the majority underwent reductional segregation due to the presence of chiasmata and tension exerted across homologs, as reported previously (Miyazaki et al, 2017) (Fig 2B). Moreover, 20% of the cells with reductional segregation at meiosis I underwent nondisjunction of sister chromatids in meiosis II, suggesting that 40% of the reductional population underwent random segregation at meiosis II which is originated from loss of cohesion (a defect in cohesion protection) in anaphase I. In contrast, *rec8-15A* cells showed nearly normal chromosome segregation in both meiosis I and II (Fig 2B).

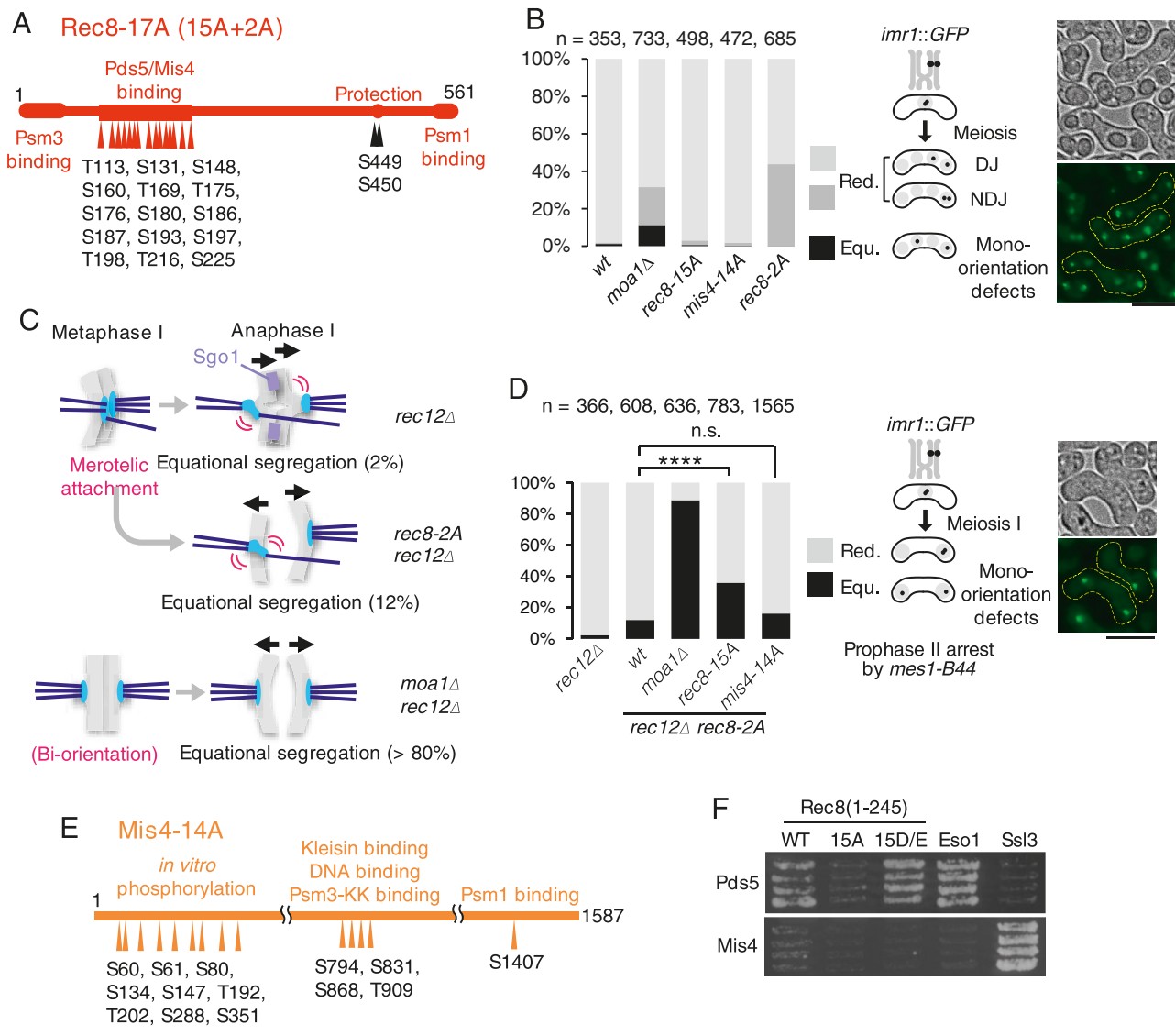

**Figure 2. Phosphorylation of Rec8 is required for mono-orientation of kinetochores in meiosis I.**
**(A)** Schematic representation of the Rec8 protein. Red arrowheads indicate 11 polo-kinase consensus sites and four non-consensus S/T in the domain including putative Mis4- or Pds5-binding sequences of Rec8 (Fig S1). Two black arrowheads indicate two phosphorylation sites required for protection of cohesin by Sgo1-PP2A. **(B)** The segregation pattern of *imr1::GFP* marked on one homolog was monitored in cells after the meiosis II division. DJ: disjunction, meaning proper sister chromatid segregation in meiosis II. NDJ: nondisjunction, meaning sister chromatids move to the same pole in meiosis II, the defect originated from loss of cohesion during anaphase I. (n) Cell number used for assay. Representative bright-field (BF) and fluorescent images (maximum z-dimension projections) of the assay are shown. Scale bar, 10 *μ*m. **(C)** Representative univalent behavior in each mutant is shown schematically. Cohesion at the core centromere but not at the pericentromeric region is released at anaphase I, accounting for kinetochore splitting of univalents at this stage. Gray rod, chromatid; light blue, kinetochores; blue line, spindle; black arrowheads, the direction of chromatid movement. **(D)** The segregation pattern of *imr1::GFP* marked on one homolog was monitored in cells after the meiosis I division (arrested by *mes1-B44*). (n) Cell number used for assay. n.s., not significant; ****P < 0.0001, one-way ANOVA with Bonferroni's multiple comparisons test. Scale bar, 10 *μ*m. **(E)** Schematic representation of the Mis4 protein. The arrowheads indicate polo-kinase consensus sites which were mutated to alanine in Mis4-14A. **(F)** Yeast two-hybrid assay examining the interaction of the N terminus of Rec8 (1–245 a.a.) with Pds5 and Mis4. Eso1 and Ssl3 are the positive interaction controls of Pds5 and Mis4, respectively. Source data are available for this figure.

Because a mono-orientation defect at the core centromere is greatly suppressed by chiasmata and pericentromeric cohesion protection, we removed these two effects by deleting the recombination gene *rec12* (the Spo11 homolog in fission yeast) and by introducing a protection defective mutation in Rec8 (*rec8-2A: S449A, S450A*) (Ma et al, 2021) (Fig 2B). In *rec12Δ* cells, merotelic attachment of mono-oriented (fused) kinetochores often causes bi-orientation

of homologs at metaphase I but sister chromatids largely fail to separate in the following anaphase because of protection of cohesion at the pericentromere, resulting in only rare (<2%) equational segregation (Sakuno et al, 2011) (Fig 2C). However, in *rec12Δ rec8-2A* cells, the separation increases to 12% because of the loss of protection (Fig 2C and D). Deletion of *moa1* in this background, by abolishing mono-orientation, leads to >80% of cells showing

equational segregation. Remarkably, introducing the *rec8-15A* mutation into *rec12Δ rec8-2A* cells increased equational segregation to 36% (Fig 2D), suggesting that phosphorylation at some or all 15S/T sites in Rec8 is contributing at least partly to establishing mono-orientation, most likely by promoting cohesion at the core centromeres.

Next, we asked if phosphorylation of Mis4 might contribute to mono-orientation by facilitating cohesin conversion from the Mis4-bound into Pds5-bound (Fig 1A). Structural analyses indicate that the middle and C-terminal region of Mis4 directly interact with the kleisin subunit, Psm1, Psm3-K105K106 and DNA, and play a key role in cohesin loading and translocation (Kikuchi et al, 2016; Collier et al, 2020; Higashi et al, 2020; Shi et al, 2020) (Fig 1A, square). We identified five polo-kinase consensus sequences in these domains (Fig 2E). Furthermore, we examined the in vitro phosphorylation by Plo1 using recombinant Mis4 protein and found that the N terminus of Mis4 is preferentially phosphorylated by Plo1 (Fig S2A–C). This domain contains nine polo-kinase consensus sequences (Fig S2D). Consequently, we introduced alanine substitutions in totally 14 polo-kinase consensus sequences in Mis4 (Fig 2E). However, *mis4-14A* cells showed only 16% equational segregation compared with WT cells (12%) in the *rec12Δ rec8-2A* background, implying that few defects in mono-orientation were caused by *mis4-14A* (Fig 2D).

To delineate the interaction of Rec8 with Mis4 or Pds5, we examined their interactions by two-hybrid assay using the N-terminal fragment of Rec8 (1–245 a.a.) that includes the Mis4/Pds5-binding surface. Rec8-N indeed interacted with Pds5 but little with Mis4. Remarkably, the interaction between Rec8 and Pds5 was reduced in non-phosphorylatable Rec8-15A but robust in Rec8-15D/E which carries phosphomimetic replacements (Fig 2F). Although these results are consistent with the hypothesis that phosphorylation of Rec8 by Moa1-Plo1 facilitates converting Mis4-bound Rec8 cohesin complex into Pds5-bound (Fig 1A), further detailed analysis of each phosphorylation site of Rec8 and Mis4/Pds5 might be required for drawing firm conclusion.

## Phosphorylation in the kleisin-Psm3 exit/entry gate

Mis4 entraps DNAs near the kleisin-Psm3 gate (Collier et al, 2020; Higashi et al, 2020; Shi et al, 2020). Once Psm3-K105K106 acetylation takes place during DNA replication (Moldovan et al, 2006), Mis4 is readily replaced by Pds5 because Pds5 but not Mis4 binds acetylated Psm3 (Petela et al, 2021). Therefore, it is possible that the kleisin-Psm3 gate and nearby Psm3-K105K106 acetylation domain might also be the target of regulation by Moa1-Plo1 (Fig 1A, square). We examined if Psm3 is phosphorylated by Plo1 in vitro and found phosphorylation of T182 in the N terminus and S1001 in the C terminus, both locate in the coiled-coil region of the DNA exit gate (Figs 3A and S3A–E), which directly interacts with the N-terminal domain of kleisin (kleisin's NTD) (Figs 1A and 3B). Indeed, cells expressing Psm3-2A (alanine substitution at T182 and S1001) showed mono-orientation defects (20%) albeit mildly, compared with WT (12%) (Fig 3C), suggesting that phosphorylation of these sites might contribute to establishing cohesion at the core centromeres. We analyzed phosphomimetic mutants *psm3-T182E* or *psm3-S1001D*. Although neither mutation showed defects in

meiotic chromosome segregation in *rec12⁺* cells, *psm3-ED* (*T182E* and *S1001D*) caused equational segregation in >70% cells (Fig 3D). This is a reminiscent of *rec8Δ* which also causes equational rather than random segregation at meiosis I because the residual Rad21 cohesin sustains centromeric cohesion (Yokobayashi et al, 2003). We assayed sister chromatid cohesion in prophase I arrested cells by monitoring GFP fluorescence associated with a *lacO* array integrated at the centromere of one homolog of chromosome I (*imr1*::GFP) and at the arm region of one homolog of chromosome II (*cut3*::GFP). The results indicate that cohesion defects in *psm3-ED* cells is a reminiscence of *rec8Δ* (Fig 3E). Thus, we thought that Rec8 cohesin function was impaired in *psm3-ED* cells. Accordingly, immunofluorescence examination indicated that the localization of Rec8 was largely lost in meiotic *psm3-ED* cells (Fig 3F). Furthermore, Western blot assay indicated that the amount of Rec8 protein was reduced in *psm3-ED* cells but not in WT and *psm3-2A* cells (Fig S4), suggesting that Rec8 protein is not stable in *psm3-ED* cells. However, *psm3-ED* did not affect Rad21 localization during vegetative growth and cells were not sensitive to thiabendazole (TBZ) (Fig 3G and H), indicating that Rad21 cohesin function is intact in *psm3-ED* cells. These results are consistent with the scenario that phosphorylation at Psm3-T182 and -S1001 might open the DNA-entry/exit gate only when it is formed by Rec8's NTD but not by Rad21's NTD. We speculate that the Psm3 phosphorylation at the gate may play a role in transiently loosening Rec8-Psm3 gate to facilitate establishment of cohesion at the core centromere.

## A phosphorylation located adjacent to the Psm3 acetylation loop

The in vitro Plo1 phosphorylation assay further identified Psm3-S110 fitting with the polo-kinase consensus sequence (Figs 4A and S3C). Mutagenesis revealed that *psm3-S110A* (alanine substitution at the S110) causes mono-orientation defects (50%) compared with WT (12%) in meiotic cells (Fig 4B). Because Psm3-S110 positions at the anti-parallel β-sheets structure that extrudes a peptide loop containing K105 and K106 residues of Psm3 (Psm3's KKD loop) (Kouznetsova et al, 2016), the phosphorylation at S110 might structurally affect Psm3's KKD loop, which plays a key role in cohesin loading and releasing function. Because *psm3-S110D* (phosphomimetic substitution at the S110) also causes mono-orientation defects (Fig 4B), we assume that the phosphorylation may occur only transiently. Importantly, either mutation does not affect the mitotic function of cohesin as the mutants show no defects in growth on a TBZ plate (Fig 4C). These results suggest that the phosphorylation at Psm3-S110 specifically regulates Rec8 cohesin at centromeres most likely depending on Moa1-Plo1. Moreover, even in meiosis, chromosome segregation in meiosis I and meiosis II are mostly normal in these mutants (Fig 4D), indicating that in contrast to the core centromere, pericentromeric cohesion, which is essential for correct meiosis II, is largely intact. Mono-orientation defects of *psm3-S110A* and *psm3-S110D* are exposed only in the *rec12Δ rec8-2A* background. Intriguingly, in budding yeast and human Smc3, the polo-kinase consensus sequence is also found at the identical parallel β-sheets but in the opposite side (Fig S5A and B), which might similarly affect Smc3's KKD loop structure if the residue is phosphorylated by polo-like kinase.

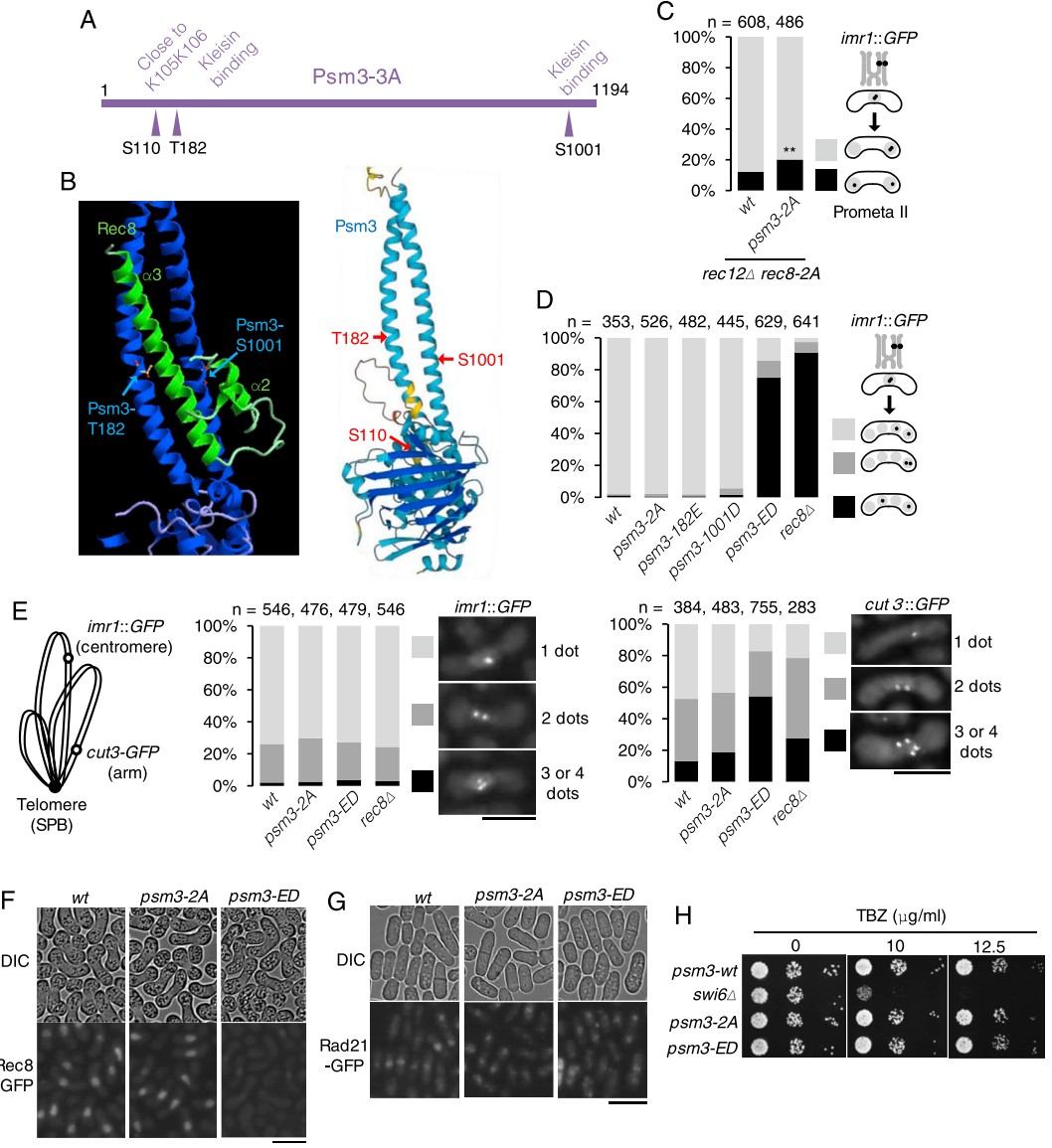

**Figure 3. Phosphorylation in the kleisin-Psm3 exit/entry gate regulates the Rec8 complex.**
**(A)** Schematic representation of the Psm3 protein. The arrowheads indicate polo-kinase consensus sites which were mutated to alanine in Psm3 (Fig S3). **(B)** Modelled structure of the Psm3-Rec8 gate domain from budding yeast Smc3-Scc1 (4UX3) RCSB Protein Data Bank (left) and AlphaFold prediction for *S. pombe* Psm3 (right). **(C)** The segregation pattern of *imr1*::GFP marked on one homolog was monitored in cells after meiosis I division (arrested by *mes1-B44*). (n) Cell number used for assay. **P < 0.01, unpaired t test. **(D)** The segregation pattern of *imr1*::GFP marked on one homolog was monitored in cells after meiosis II division. (n) Cell number used for assay. **(E)** *imr1*::GFP and *cut3*::GFP were observed in prophase I (*mei4Δ*-arrested) $h^{90}$ cells. The number of dots per nucleus is shown with photos of representative nuclei. (n) Cell number used for assay. Scale bars, 10 μm. **(F)** Rec8-GFP signals were observed in indicated prophase I–arrested cells. Scale bar, 10 μm. **(G)** Rad21-GFP signals were observed in indicated proliferating cells. Scale bar, 10 μm. **(H)** Serial 10-fold dilutions of the indicated cells were spotted on YE plates including and lacking thiabendazole and grown at 30°C for 2 d.
Source data are available for this figure.

To explore the relation among the phosphorylation on Rec8 and Psm3, we combined *rec8-15A* and *psm3-3A* (*S110A, T182A,* and *S1001A*), and examined mono-orientation defects in *rec12Δ rec8-2A* background. Accordingly, *rec8-15A psm3-3A* double mutant showed more defects in mono-orientation than either *rec8-15A* or *psm3-3A* mutant (Fig 4E), suggesting that the phosphorylation on Rec8 and Psm3 cooperatively act to establish cohesion at the core centromeres. When *wpl1⁺* is deleted in the *rec8-15A*, *psm3-3A*, and *rec8-15A psm3-3A* mutants, mono-orientation defects were suppressed in

*psm3-3A* and *rec8-15A psm3-3A* mutants but not in *rec8-15A* (Fig 4E). These results support the notion that the *psm3-3A* mutation locating near the Rec8-Psm3 gate and Psm3-KKD-loop enhanced the cohesin release pathway, which depends on Wpl1.

In conclusion, we identified several Plo1 phosphorylation sites in the cohesin subunits, Rec8 and Psm3, non-phosphorylatable mutations of which cause defects in mono-orientation in meiosis I. We argue that the phosphorylation of Rec8 and Psm3 at the core centromeres is at least partly mediated by kinetochore

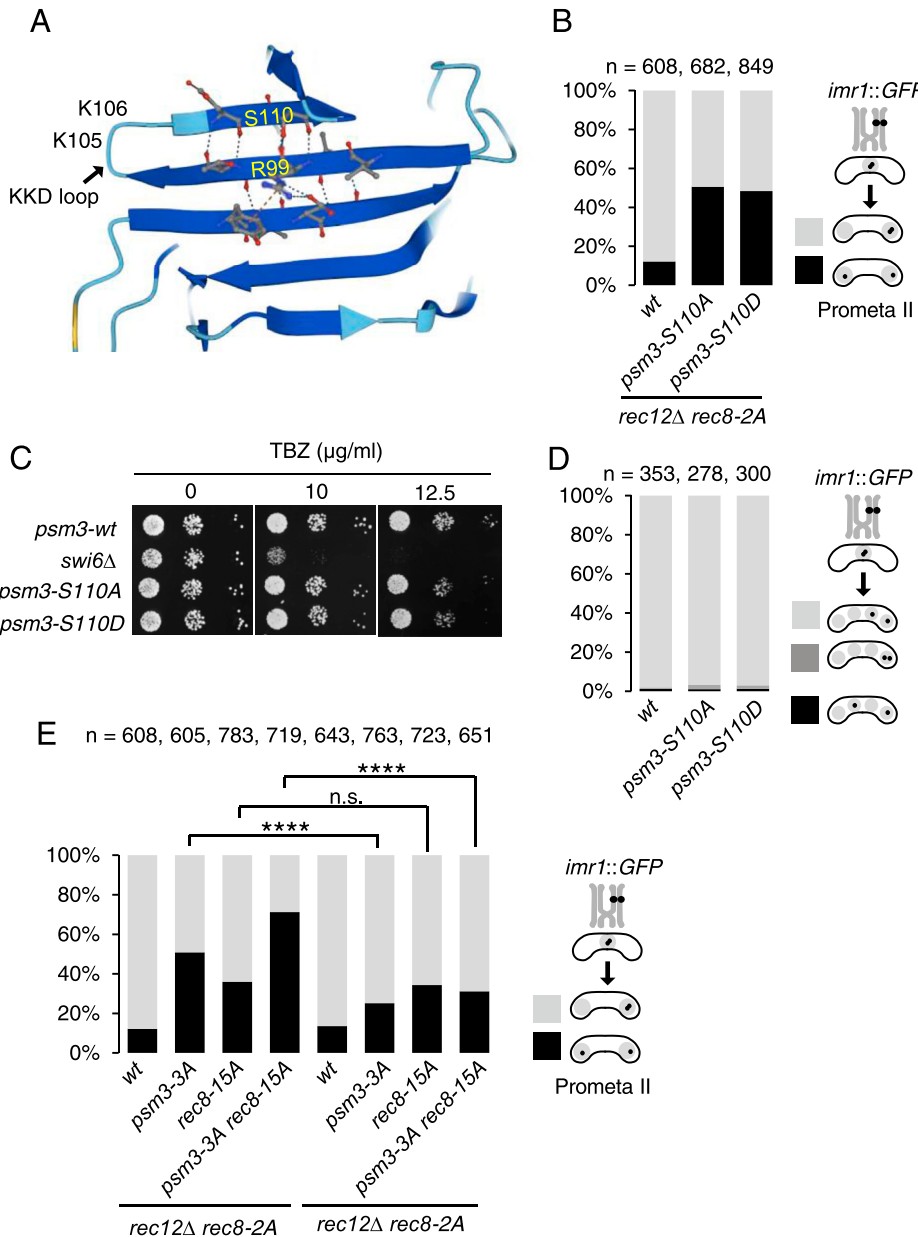

**Figure 4. Phosphorylation of Psm3 and Rec8 is required for mono-orientation.**
**(A)** Schematic representation of Psm3's KKD loop and the parallel β-sheet. **(B)** The segregation pattern of *imr1*::GFP marked on one homolog was monitored in cells after the meiosis I division (arrested by *mes1-B44*). (n) Cell number used for assay. **(C)** Serial 10-fold dilutions of the indicated cells were spotted on YE plates including and lacking thiabendazole and grown at 30°C for 2 d. **(D)** The segregation pattern of *imr1*::GFP marked on one homolog was monitored in cells after the meiosis II division. (n) Cell number used for assay. **(E)** The segregation pattern of *imr1*::GFP marked on one homolog was monitored in cells after meiosis I division (arrested by *mes1-B44*). n.s., not significant; ****$P < 0.0001$, one-way ANOVA with Bonferroni's multiple comparisons test. (n) Cell number used for assay.
Source data are available for this figure.

localized Moa1-Plo1 that regulates Rec8 cohesin complexes to establish cohesion required for mono-orientation (Yokobayashi & Watanabe, 2005; Sakuno et al, 2009). Although previous studies suggest that Rec8 cohesin dependent cohesion at the core centromere is important for mono-orientation in various organisms including plants and animals (Watanabe, 2012; Kim et al, 2015; Ogushi et al, 2021), it has been elusive in budding yeast, the only of the studied organisms having a point centromere. However, a recent study in budding yeast suggests that cohesin indeed plays an important role in establishing mono-orientation (Barton et al, 2022). Thus, we anticipate that the molecular mechanisms of meiosis-specific kinetochore regulation revealed in fission yeast might be conserved in a wide range of eukaryotic organisms, including human.

# Materials and Methods

### *Schizosaccharomyces pombe* strains and media

Unless otherwise stated, all media and growth conditions were as described previously (Moreno et al, 1991). Complete medium (YE), minimal medium (EMM), minimal medium without nitrogen source (EMM-N) and sporulation-inducing media (SPA) were used. All strains used in this study are described in Table S1. Deletion of *rec8+*, *moa1+*, *rec12+*, *mei4+*, and *wpl1+*; tagging of *rec8+*, *rad21+*, and *pds5+* by GFP; and tagging of *mis4+* by PK were carried out according to the PCR-based gene targeting method for *S. pombe* using the kanMX6 (kanR), hphMX6 (hygR), natMX6 (natR), bsdMX6 (bsdR), and aurMX6 (aurR)

genes as selection markers. We used *imr1*::GFP and *cut3*::GFP markers to monitor cohesion of sister chromatids (Tomonaga et al, 2000; Sakuno et al, 2009). To introduce mutants in *rec8*[+], *psm3*[+], and *mis4*[+] genes, we cloned fragments of these genes into plasmid, respectively, and mutagenized using whole plasmid mutagenesis, correct mutagenesis was confirmed by PCR and/or DNA sequencing.

## Synchronous cultures of fission yeast meiotic cells

For meiosis microscopic observation, logarithmically growing cells were collected, suspended in 20 mg/ml Leucine, spotted on sporulation-inducing medium (SPA), and incubated at 28°C. For chromosome segregation assay, *imr1*-GFP was observed in the *mes1*[+] or *mes1-B44* mutant that arrests at prophase II (after 16 h). For sister chromatid cohesion assay, *imr1*-GFP and *cut3*-GFP were observed in *mei4Δ* mutant that arrests at prophase I (after 24 h).

## In vitro phosphorylation assay of Rec8, Mis4, and Psm3

The recombinant GST or MBP fusion proteins (Rec8, Mis4, and Psm3) were produced in *Escherichia coli* using the pGEX4T-2 and pGEX4T-1 vector (GE Healthcare). *E. coli* was lysed in lysis buffer (40 mM Tris–HCl [pH 7.5], 0.5 mM EDTA, 150 mM NaCl, and 0.5% Triton X-100), and GST fusion proteins were purified on glutathione–Sepharose (GE Healthcare) and MBP fusion proteins were purified by amylose resin (NEB). Recombinant Plo1 kinase and substrates were incubated with kinase buffer (50 mM Tris–HCl [pH 8.0], 150 mM NaCl, 10 mM $MgCl_2$, 0.5% Triton X-100, and 5 mM dithiothreitol) at 30°C for 30 min in the presence of ($\gamma$-$^{32}$P) ATP or cold ATP. The incorporation of the radioactive phosphate group was visualized by means of autoradiography, and protein loading was analyzed by staining with Coomassie brilliant blue. Phosphopeptides were analyzed by liquid chromatography coupled with tandem mass spectrometry (LC–MS/MS).

## In vivo phosphorylation assay of Rec8

Protein bands were excised and in-gel digested using trypsin. The peptides were analyzed with an Orbitrap-Fusion Lumos mass spectrometer coupled to an UltiMate 3000 HPLC equipped with an EASY-Spray nanosource (Thermo Fisher Scientific). Raw data were processed using MaxQuant v1.6.0.1 and searched against a UniProt extracted *S. pombe* FASTA file amended to include common contaminants, with phosphorylation (STY), acetylation (K), and methylation (KR) being selected as variable modifications. The proteingroup.txt, phosphoS-TY.txt output tables were imported into Perseus software for further processing. All intensity values were log2 transformed.

## Fluorescence microscopy

All cell fluorescence microscopy was performed using a Nikon ECLIPSE Ti2-E inverted microscope with photometrics PRIME 95B camera. This microscope was controlled by NIS-Elements software. Thirteen z sections (spaced by 0.4 μm each) of the fluorescent images were converted into a single two-dimensional image by maximum intensity projection. Image J software (NIH) was used to adjust brightness and contrast and to render maximum projection images.

## Two-hybrid assays

The yeast two-hybrid system (BD) was used according to Clontech's instructions. We amplified the ORFs of *pds5*[+] and *mis4*[+] genes by PCR and cloned them into pGBK-T7, a Gal4 DNA-binding domain–based bait vector, amplified *rec8*[+], *rec8-15A*, *rec8-15D/E*, *eso1*[+], and *ssl3*[+] and cloned into pGAD-T7, a Gal4 activation domain–based prey vector. Y2HGold and Y187 strains were used for transformation of pGBK-T7 and pGAD-T7, respectively. SD-tryptophan and SD-leucine plates were used as selective medium. Positive transformants of Y2HGold and Y187 were mixed on YPD plate to get diploid cells and streaked on SD-Trp-Leu plate to get single colonies; checked on nutrition-restricted plates (SD-Trp-Leu-His-Ade).

## Serial dilution analyses

TBZ was dissolved in DMSO as a stock solution at 20 mg/ml and stored at room temperature. Sterilized YE agar medium (115°C, 20 min) was cooled down (about 60°C) and added with TBZ to prepare the plates. For serial dilution plating assays, 10-fold dilutions of a mid-log phase culture (OD660 = 0.3~0.6) were plated on the indicated medium and grown for 2 d at 30°C.

## Chromatin immunoprecipitation assays

For the chromatin immunoprecipitation analysis, we used $h^+/h^-$ diploid cells. *mei4Δ* was used to arrest at prophase I. Cells were grown in YE liquid to a density of $1 \times 10^7$ cells/ml at 30°C, then resuspended in EMM liquid medium including $NH_4Cl$ (EMM + N) and grown to a density of $0.5 \times 10^7$ cells/ml at 30°C, then resuspended in EMM medium lacking $NH_4Cl$ (EMM – N) at a density of $1 \times 10^7$ cells/ml for 9 h at 30°C. ChIP assays were carried out as described previously (Yokobayashi & Watanabe, 2005). The sequences of primers used are as follows: *cnt* primers (5′-ATCTCATTGCTATTTGGCGAC-3′, 5′-GCGTTTCTTCGGCGAAATGC-3′), *imr* primers (5′-CCTTTACTG-GAAAATTGTCG-3′, 5′-GCTGAGGCTAAGTATCTGTT-3′), *dg* primers (5′-TTTTCAGCGAGACATGTACC-3′, 5′-TCATAAAGCAACACTGGGTG-3′), *dh* primers (5′-TGAATCGTGTCACTCAACCC-3′, 5′-CGAAACTTTCAGATCTCGCC-3′), *zfs1* primers (5′-CCGGTTGAAAGGAATAGACT-3′, 5′-TTTCTTGCCTGA-GATACCGT-3′), and *mes1* primers (5′-CGAAGGCTACTTTCATGCCA-3′, 5′-CGTACATTCAGACTGTTGAAC-3′). After getting the qPCR results, we divided the value of *cnt*, *imr*, *dg*, and *dh* by the mean value of *zfs1* and *mes1* because the *zfs1* and *mes1* regions are basically free of cohesin and the detected values belong to background (Yokobayashi & Watanabe, 2005; Goto et al, 2017). The final values were obtained by subtracting the value calculated from control strains without the tag.

## Statistical analysis

All the data replicates were applied and analyzed using GraphPad Prism version 9.5.1 (GraphPad Software). To estimate the significant differences between two groups, *t* test was performed. To estimate the significant differences between ≥3 groups, one-way ANOVA was performed.

# Supplementary Information

# Acknowledgements

We thank Silke Hauf for critically reading the manuscript, Andrew Jones for mass spectrometry analysis, Jian Chen for general support, and the Yeast Genetic Resource Center (YGRC) for yeast strains and plasmids. We also thank Takafumi Ishihara for initial analyses of Psm3 phosphorylation. This work was supported by the National Key Research and Development Program of China (2017YFC1600403), the National Natural Science Foundation of China (Key Program, 31830068), and Entrepreneurial and innovative talent of Jiangsu (JSSCRC2021495).

### Author Contributions

Y Liu: data curation, software, formal analysis, and investigation.
Y Min: data curation and methodology.
Y Liu: methodology.
Y Watanabe: conceptualization, data curation, formal analysis, supervision, funding acquisition, validation, investigation, methodology, and writing—original draft.

### Conflict of Interest Statement

The authors declare that they have no conflict of interest.

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
