## [Reviewer comments · Life Science Alliance]

Life Science Alliance

Phosphorylation of Rec8 cohesin complexes regulates mono-orientation of kinetochores in meiosis I

Yu Liu, Yu Min, Yongxin Liu, and Yoshinori Watanabe

DOI: <https://doi.org/10.26508/lsa.202302556>

Corresponding author(s): Yoshinori Watanabe, Jiangnan University

Review Timeline:

Submission Date:	2023-12-23
Editorial Decision:	2024-01-29
Revision Received:	2024-02-08
Editorial Decision:	2024-02-22
Revision Received:	2024-02-27
Accepted:	2024-02-27

Transaction Report:

January 29, 2024

Re: Life Science Alliance manuscript #LSA-2023-02556

Prof. Yoshinori Watanabe
Jiangnan University
Science Center for Future Foods
1800 Lihu Avenue
Wuxi, Jiangsu 214122
China

Dear Dr. Watanabe,

Thank you for submitting your manuscript entitled "Phosphorylation of Rec8 cohesin complexes regulates mono-orientation of kinetochores in meiosis I" to Life Science Alliance. The manuscript was assessed by expert reviewers, whose comments are appended to this letter. We invite you to submit a revised manuscript addressing the Reviewer comments.

Thank you for this interesting contribution to Life Science Alliance. We are looking forward to receiving your revised manuscript.

Sincerely,

B. MANUSCRIPT ORGANIZATION AND FORMATTING:

Reviewer #1 (Comments to the Authors (Required)):

The cohesin complex is essential for keeping the sister chromatids together from S-phase to Anaphase (and Anaphase II in meiosis). It comprises four subunits (Psm3, Psm1, Rad21, and Psc3 in *S. pombe*) and is highly conserved from yeast to humans. The kleisin subunit (Rad21) is replaced by Rec8 during meiosis, and the significance of this replacement for meiosis-specific functions is poorly understood. The meiotic kinetochore protein Moa1 interacts with a polo-like kinase Plo1 to phosphorylate several residues on Rec8. Previously, the same group demonstrated the Plo1-mediated phosphorylation of Rec8-S450 and its significance in protecting centromeric cohesion (Ma et al. 2021). In this manuscript, Liu et al. nicely demonstrate the role of Plo1-mediated phosphorylation of cohesin subunits (Rec8 and Psm3) in regulating sister-kinetochore mono-orientation during meiosis I. Interestingly, they have thoughtfully used the structural information available for the cohesin subunits and their regulators to decipher the role of Rec8 and Psm3 phosphorylation in executing mono-orientation. The manuscript is well-written, and the experiments are well-designed and nicely interpreted. This manuscript is suitable for publication after minor corrections, as listed below:

- 1) Page 4: The authors have stated, "Accordingly, immunofluorescence examination indicated that Rec8 was largely lost from chromatin in meiotic psm3-ED cells (Fig. 3F)." Based on the immunofluorescence analysis, it is difficult to say whether the chromatin-bound Rec8 was lost or the Rec8 protein expression was reduced. I suggest authors show this data by chromatin spread (or ChIP) to confirm that chromatin-bound Rec8 is reduced. The presented immunofluorescence data (Fig. 3F) suggest that the Rec8 protein expression is reduced. Can the author clarify this issue?
- 2) Page 5: The authors have discussed data with reference to Fig. 4F. However, Fig. 4F does not exist.
- 3) Page 6: Method for ChIP assay: The authors have used the values for *zfs1* and *mes1* to normalize the ChIP data. It would be helpful for the readers if the authors could briefly mention the rationale for selecting *zfs1* and *mes1* for normalization.
- 4) Throughout the manuscript, in several places, the authors have not mentioned specific numbers for the % cells with equational/reductional chromosome segregation. It will be helpful for the readers if the authors can mention specific numbers. E.g. Page 3, "However, *mis4-14A* showed few defects in mono-orientation in meiosis I (Fig. 2D)." Instead, write, "However, *mis4-14A* showed ~19% cells with equational segregation, suggesting a defect in mono-orientation in meiosis I (Fig. 2D)."
- 5) Mention the full form of TBZ in the Material and Methods section.
- 6) Page 4: The authors should mention any reference or the rationale for the role of KKD loop in the cohesin function.

Reviewer #2 (Comments to the Authors (Required)):

The manuscript by Yu Liu and co-authors addresses how mono-orientation of sister kinetochores in meiosis I is regulated by the phosphorylation of cohesin subunits in a Plk1-kinase dependent manner, using fission yeast as a model system. In the first part of the manuscript, the authors show evidence that Rec8 has to be phosphorylated in order to convert Mis4-dependent cohesin loading into Pds5-dependent cohesin establishment at the centromere. They propose that phosphorylation of Rec8 by Plo1 (Plk1 in *S. pombe*) is necessary for cohesin establishment at the centromere and thus mono-orientation. Then, they examine adjacent phosphorylation at the kleisin-Psm3 exit/entry gate by Plo1, and conclude as to those sites being important for mono-orientation as well.

Overall, the manuscript is not well written, and hard to follow. It is not clear what hypothesis is being tested and how conclusions are reached. Incoherences in describing previous literature makes it nearly impossible to follow the logic of the experiments, or how the authors reach their conclusions. (for example page 3, top paragraph, description of the *moa1* deletion: "20 % of cells undergoing a reductional division show chromosome non-disjunction in meiosis II, indicative of a defect in cohesion protection"- either there is a defect in cohesin protection, with precocious sister separation, or there is chromosome non-disjunction, which is

the opposite).

Main points:

- 1) the authors use phosphomimicking and non-phosphorylatable mutants without once showing that any of those sites are indeed phosphorylated in vivo.
- 2) the authors state that those sites are specifically phosphorylated by Moa1-Plo1 without any data showing that Plo1 indeed needs to be in complex with Moa1 for the phosphorylation of these sites.
- 3) Statistical tests (nearly all figures) and controls are missing, or not sufficiently described. (for example, Figure 2, for the conclusion of the figure statistics in Figure 2D are essential, the control *rec12delta* alone is missing, etc).
- 4) Which method was used to determine whether a Plo1 site is phosphorylated in vitro, or not? How was this decided? (Fold changes? By how much?)
- 5) The authors should explain why the non-phosphorylatable mutant of Mis4 (Mis4-14A) was expected to affect mono-orientation. Do they expect Mis4 to be phosphorylated so that cohesin loading is converted to cohesion establishment? This needs to be put into perspective with Figure 1B and then the authors should provide an interpretation of their results (eg, why no phenotype is observed).
- 6) The authors analyzed the phosphomimetic mutants *psm3-T182E* or *psm3-S1001D*. (neither mutation showed defects in meiotic chromosome segregation in *rec12+* cells, but *psm3-ED* (T182E and S1001D) caused equational segregation in >70% cells.) This should be part of Figure 3D.
- 7) Can the authors explain why *psm3-ED* has a stronger phenotype than *delta Rec8*? (statistics are missing here, too)
- 8) Figure legends not clear, statistical tests not explained (Figure 1), abbreviations not defined (eg DJ and NDJ in Figure 2B)
- 9) Figure 2F is a key result, should be quantified from several repeats.
- 10) Can the authors explain why *psm3-S110A* and *psm3-S110D* have the same phenotype, if Plo1 phosphorylation on this site is important in vivo for centromeric cohesion and mono-orientation?

Reviewer #3 (Comments to the Authors (Required)):

Review on the LSA-manuscript 2023-02556 entitled "Phosphorylation of Rec8 cohesin complexes regulates mono-orientation of sister kinetochores in meiosis I" by Yu Liu, Yu Min, Yongxin Liu and Yoshinori Watanabe

Meikin/Moa1 does double duty in meiosis I. It mediates Sgo1-PP2A-dependent protection of pericentromeric cohesin from cleavage by separase, thereby preventing premature sister chromatid separation in anaphase I, and it mediates fusion of sister kinetochores and their consequent mono-orientation on the meiosis I spindle, thereby ensuring proper segregation of homologous chromosomes. Meikin/Moa1 is known to associate with polo-like kinase/Plo1. Previous studies from the Watanabe lab reported how Moa1-Plo1-dependent phosphorylation of Rec8 on S450 (and of KNL1/Spc7) lead to efficient protection of pericentromeric cohesion. It is known that cohesion of core centromeres is required for sister kinetochore fusion but how the Moa1-Plo1 complex promotes mono-orientation remains largely enigmatic. In the manuscript at hand, Watanabe and co-workers report the following:

- 1) Using kinase assays and mass spectrometry, they identified sites within Rec8, Mis4 and Psm3 that can be phosphorylated by Plo1 in vitro. The authors then mutated these sites plus additional putative ones, which map the consensus for polo substrates. Subsequently, the corresponding variants were functionally characterized in *S. pombe* meiosis.
- 2) Changing 15 Ser/Thr residues in Rec8 to Ala caused a partial defect in mono-orientation but seemed to leave pericentromeric cohesion protection largely unaffected.
- 3) Using acidic residues to mimic constitutive phosphorylation of Psm3 at T182 and S1001 resulted in an almost complete loss of Rec8-cohesin function. Because T182 and S1001 map to the neck of Psm3's coiled coil which is in contact with Rec8's NTD, the authors speculate that phosphorylations at these positions might lead to opening of the Rec8-Psm3 gate.
- 4) Psm3-T182A, S1001A exhibited a weak and Psm3-S110A (or D) a more pronounced mono-orientation defect. Interestingly, the latter position maps close to the acetylation sites of Psm3. Again, both variants seem to retain their cohesive function. Combining Rec8-15A with Psm3-3A aggravates the mono-orientation defect, which is consistent with a role of Rec8- and Psm3-phosphorylation in cohesion at the core centromeres.

While Watanabe and co-workers had previously isolated a Rec8 variant with a selective defect in pericentromeric cohesion, they now identify cohesin variants that seem to have a selective defect in cohesion of core centromeres. The shown experiments are

carefully executed and the presented data of high quality. However, whether the identified residues are phosphorylated in vivo and - if yes - whether the phosphorylations are put in place by Moa1-associated Plo1 remains speculative.

The authors claim that their ChIP- and 2-hybrid results (Figs. 1B and 2F) together "support the hypothesis that phosphorylation of Rec8 by Moa1-Plo1 facilitates converting Mis4-bound Rec8 cohesin complex into Pds5-bound, thus promoting establishment of cohesion." This is an overinterpretation. The authors should at least repeat the ChIP-assay with a strain that relies on Rec8-15A or tone down their statement.

Minor points:

- A sentence from the introduction is contradicting one in the results: "Cohesion is established through exchanging the loader complexes with a key cohesin accessory factor, Pds5, which recruits Eso1, a cohesin acetylase (Vaur et al. 2012; Goto et al. 2017)." versus "Once Psm3-K105K106 acetylation takes place, Mis4 is replaced by Pds5 because Pds5 but not Mis4 binds acetylated Psm3 (Petela et al. 2021)."
- There is no reference to Fig. 3D in the text.
- "non-phosphorylatable" is not a proper word.
- The first sentence of the introduction could lead readers to think that cohesion is mediated by cohesin only in fission yeast. I recommend changing it to: "Sister chromatid cohesion of chromosomes is carried out by a multi-subunit complex, called cohesin, which in fission yeast comprises two SMC (structural maintenance of chromosome) family proteins, Psm1 and Psm3, a kleisin Rad21 subunit and an accessory subunit Psc3.
- It should be mentioned that separase "cleaves Rad21 along the entire chromosome" only in yeast.
- There are two typos on page 2: "Mis4/Psd5".
- The expression "phosphorylation in the either or all 15S/T sites" needs improvement.
- We identified five polo-kinase consensus sequenceS in these domains (Fig. 2E).
- Bottom of Fig. 2C: Shouldn't the genotype include rec8-2A and the frequency of equational segregation lowered from 100% to >80% to match the data shown in 2D?

Response to the referees,

We thank the referees for their valuable comments. To address the referees' comments, we have revised our manuscript. We addressed all comments raised by the referees and our responses are listed below.

(Bold letters are our responses)

Reviewer 1 Reviewer's Comments to Author

..... The manuscript is well-written, and the experiments are well-designed and nicely interpreted. This manuscript is suitable for publication after minor corrections, as listed below:

1) Page 4: The authors have stated, "Accordingly, immunofluorescence examination indicated that Rec8 was largely lost from chromatin in meiotic *psm3-ED* cells (Fig. 3F)." Based on the immunofluorescence analysis, it is difficult to say whether the chromatin-bound Rec8 was lost or the Rec8 protein expression was reduced. I suggest authors show this data by chromatin spread (or ChIP) to confirm that chromatin-bound Rec8 is reduced. The presented immunofluorescence data (Fig. 3F) suggest that the Rec8 protein expression is reduced. Can the author clarify this issue?

We examined Rec8 protein levels by Western blotting, revealing that the amount of Rec8 protein decreased in *psm3-ED* cells but not in wild-type and *psm3-2A* cells (new Fig S4). Therefore, we changed the description to "Accordingly, immunofluorescence examination indicated that the localization of Rec8 was largely lost in meiotic *psm3-ED* cells (Fig 3F). Further, the Western blotting experiments indicated that the amount of Rec8 protein was reduced in *psm3-ED* cells but not in wild-type and *psm3-2A* cells, suggesting that Rec8 protein is not stable in *psm3-ED* cells. (Fig. S4)."

2) Page 5: The authors have discussed data with reference to Fig. 4F. However, Fig. 4F does not exist.

We are sorry for this mistake. Fig 4F was changed to Fig 4E.

3) Page 6: Method for ChIP assay: The authors have used the values for *zfs1* and *mes1* to normalize the ChIP data. It would be helpful for the readers if the authors could briefly mention the rationale for selecting *zfs1* and *mes1* for normalization.

According to the suggestion, we rewrote the methods as below: After getting the qPCR results, we divided the value of *cnt*, *imr*, *dg* and *dh* by the mean value of *zfs1* and *mes1*, because the *zfs1* and *mes1* regions are basically free of cohesin and the detected values belong to background (ref). The final values were obtained by subtracting the value calculated from control strains without tag.

4) Throughout the manuscript, in several places, the authors have not mentioned specific numbers for the % cells with equational/reductional chromosome segregation. It will be helpful for the readers if the authors can mention specific numbers. E.g. Page 3, "However, *mis4-14A* showed few defects in mono-orientation in meiosis I (Fig. 2D)." Instead, write, "However, *mis4-14A* showed ~19% cells with equational segregation, suggesting a defect in mono-orientation in

meiosis I (Fig. 2D)."

According to the suggestion, we now rewrite as "However, *mis4-14A* cells showed only 16% equational segregation compared with wild-type cells (12%) in the *rec12Δ rec8-2A* background, implying that few defects in mono-orientation were caused by *mis4-14A* (Fig 2D)." We also added other specific numbers for the % cells with equational/reductional chromosome segregation. "In *moa1Δ* cells, a minority population (11%) of cells underwent equational segregation at meiosis I" in Fig 2B. "Indeed, cells expressing Psm3-2A (alanine substitution at T182 and S1001) showed mono-orientation defects (20%) albeit mildly, compared with wild type (12%) (Fig 3C) " in Fig 3C. "Mutagenesis revealed that *psm3-S110A* (alanine substitution at the S110) causes mono-orientation defects (50%) compared with wild type (12%) in meiotic cells (Fig 4B)." in Fig 4B.

5) Mention the full form of TBZ in the Material and Methods section.

We improved description of materials and methods as below: Thiabendazole (TBZ) was dissolved in DMSO as a stock solution at 20 mg/ml and stored at room temperature. Sterilized YE agar medium (115°C, 20 minutes) was cooled down (60°C) and added with TBZ to prepare the plates. For serial dilution plating assays, ten-fold dilutions of a mid-log phase culture (OD660 = 0.3~0.6) were spotted on the indicated plates and grown for 2 days at 30°C.

6) Page 4: The authors should mention any reference or the rationale for the role of KKD loop in the cohesin function.

According to the suggestion, we cited the following reference, which introduces the word 'KKD loop'.

Kouznetsova E, Kanno T, Karlberg T, Thorsell AG, Wisniewska M, Kursula P, Sjogren C, Schuler H. 2016. Sister Chromatid Cohesion Establishment Factor ESCO1 Operates by Substrate-Assisted Catalysis. *Structure* 24: 789-796.

Reviewer 2 Reviewer's Comments to Author

..... Overall, the manuscript is not well written, and hard to follow. It is not clear what hypothesis is being tested and how conclusions are reached. Incoherences in describing previous literature makes it nearly impossible to follow the logic of the experiments, or how the authors reach their conclusions. (for example page 3, top paragraph, description of the *moa1* deletion: "20 % of cells undergoing a reductional division show chromosome non-disjunction in meiosis II, indicative of a defect in cohesin protection"- either there is a defect in cohesin protection, with precocious sister separation, or there is chromosome non-disjunction, which is the opposite).

We now explain the meaning of non-disjunction in meiosis II in the text (page 3 top) as bellow: "Moreover, 20% of the cells with reductional segregation at meiosis I underwent nondisjunction of sister chromatids in meiosis II, suggesting that 40% of the reductional population underwent random segregation at meiosis II which is originated from loss of cohesin (a defect in cohesin protection) in anaphase I." Further, we add an explanation about chromosome nondisjunction in meiosis II in the figure legend to Fig 2B: "NDJ: nondisjunction, meaning sister chromatids move to the same pole in meiosis II, the defect originated from loss of cohesin during anaphase I."

Main points:

1) the authors use phosphomimicking and non-phosphorylatable mutants without once showing that any of those sites are indeed phosphorylated *in vivo*.

We showed the *in vitro* phosphorylation experiments (Fig S1-S3). Now we describe the results of mass spectrometry analysis of Rec8 phosphorylated *in vivo* (Fig S1B), indicating that at least 6 among 15 sites of Rec8 are phosphorylated *in vivo*. Because mass spectrometry analysis does not detect all phosphorylation, we mutagenized 15 phosphorylation sites in the putative Pds5 and Mis4 interaction domain.

2) the authors state that those sites are specifically phosphorylated by Moa1-Plo1 without any data showing that Plo1 indeed needs to be in complex with Moa1 for the phosphorylation of these sites.

Our previous study (Ma et al. Genes Dev 2021) shows that there is mono-orientation defect in *moa1*Δ cells, and Cnp3-Plo1 can rescue the defect, suggesting that Moa1 is important for recruiting Plo1 to localize at kinetochore and Plo1 itself plays a role in establishment of mono-orientation. In the current study, we show that at least 6 among 15 sites of Rec8 are phosphorylated *in vivo* depending on Moa1 (Fig S1B). These results support the notion that Moa1 associated or dependent Plo1 indeed phosphorylates Rec8.

3) Statistical tests (nearly all figures) and controls are missing, or not sufficiently described.

(for example, Figure 2, for the conclusion of the figure statistics in Figure 2D are essential, the control *rec12*Δ alone is missing, etc).

Accordingly we added the statistics analysis and control of *rec12*Δ in new Fig 2D.

4) Which method was used to determine whether a Plo1 site is phosphorylated *in vitro*, or not? How was this decided? (Fold changes? By how much?)

We are sorry for the lack of full explanation in Fig S1. We now revised Fig S1 by including mass spectrometric analysis of *in vivo* phosphorylation of Rec8 and provided full explanation in the legend Fig S1C; In the Rec8 (111-225 a.a.) domain covering putative Mis4 or Pds5-binding sequences, we chose eleven polo-kinase consensus N/Q/E/D-X-S/T (red) and four non-consensus S/T (blue) which were also phosphorylated by Plo1 *in vitro*. Arrows indicate that at least these sites are phosphorylated *in vivo* depending on Moa1 and Plo1.

5) The authors should explain why the non-phosphorylatable mutant of Mis4 (Mis4-14A) was expected to affect mono-orientation. Do they expect Mis4 to be phosphorylated so that cohesin loading is converted to cohesion establishment? This needs to be put into perspective with Figure 1B and then the authors should provide an interpretation of their results (eg, why no phenotype is observed).

We hypothesized that phosphorylation of Mis4 might facilitate cohesin conversion from the Mis4-bound into Pds5-bound (Fig 1A). We then thought that non-phosphorylatable Mis4 might cause defects in the conversion, thus affecting cohesion establishment and

mono-orientation. However, the mono-orientation defects were very little.

We now described in the text: Next, we asked if phosphorylation of Mis4 might contribute to mono-orientation by facilitating cohesin conversion from the Mis4-bound into Pds5-bound (Fig 1A)..... However, compared with wild-type cells (12%), *mis4-14A* cells showed 16% equational segregation, indicating that only very few defects in mono-orientation were caused by *mis4-14A* (Fig 2D).

6) The authors analyzed the phosphomimetic mutants *psm3-T182E* or *psm3-S1001D*. (neither mutation showed defects in meiotic chromosome segregation in *rec12+* cells, but *psm3-ED* (T182E and S1001D) caused equational segregation in >70% cells.) This should be part of Figure 3D.

We are sorry for this mistake. This sentence belongs to Fig 3D. We corrected it.

7) Can the authors explain why *psm3-ED* has a stronger phenotype than *delta Rec8*? (statistics are missing here, too)

We do not know the real reason for the less defects in *rec8Δ*. Notably, however, *cut3::GFP* localizes at the short arm of chromosome 2 but close to telomeres, the region where *Rad21* is abundant even in *rec8Δ* cells (Yokobayashi et al MCB 2003). Thus, it is possible that *psm3-ED* might affect *Rad21* function near telomeres, thus influencing cohesion at the *cut3::GFP* locus more severely than *rec8Δ*. We added the statistics analysis in the new Fig 3E.

8) Figure legends not clear, statistical tests not explained (Figure 1), abbreviations not defined (eg DJ and NDJ in Figure 2B)

We accept the suggestion and now add the explanation and definition in the legends.

Error bars represent SD (n = 3 PCR amplifications).

DJ: disjunction, meaning proper sister chromatid segregation in meiosis II. NDJ: nondisjunction, meaning sister chromatids move to the same pole in meiosis II, the defect originated from loss of cohesion during anaphase I.

9) Figure 2F is a key result, should be quantified from several repeats.

We repeated the experiment for several times as shown below. They are all consistent with the presented data (Fig 2F).

10) Can the authors explain why psm3-S110A and psm3-S110D have the same phenotype, if Plo1 phosphorylation on this site is important *in vivo* for centromeric cohesion and mono-orientation?

We think that the phosphorylation of psm3-S110 may occur only transiently. Because cohesion establishment is a dynamic process, we assume that fixed modification S110D or loss of it S110A may cause a negative effect on the whole process.

Reviewer 3 Reviewer's Comments to Author

..... While Watanabe and co-workers had previously isolated a Rec8 variant with a selective defect in pericentromeric cohesion, they now identify cohesin variants that seem to have a selective defect in cohesion of core centromeres. The shown experiments are carefully executed and the presented data of high quality. However, whether the identified residues are phosphorylated *in vivo* and - if yes - whether the phosphorylations are put in place by Moa1-associated Plo1 remains speculative.

We now revised Fig S1 by including mass spectrometric analysis of *in vivo* phosphorylation of Rec8 and provided full explanation in the legend. In the new Fig S1, we show that at least six *in vitro* phosphorylation sites of Rec8 are phosphorylated *in vivo* depending on Moa1 and Plo1.

The authors claim that their ChIP- and 2-hybrid results (Figs. 1B and 2F) together "support the hypothesis that phosphorylation of Rec8 by Moa1-Plo1 facilitates converting Mis4-bound Rec8 cohesin complex into Pds5-bound, thus promoting establishment of cohesion." This is an overinterpretation. The authors should at least repeat the ChIP-assay with a strain that relies on Rec8-15A or tone down their statement.

Accordingly, we toned down the statement: " Although these results are consistent with the hypothesis that phosphorylation of Rec8 by Moa1-Plo1 facilitates converting Mis4-bound Rec8 cohesin complex into Pds5-bound (Fig 1A), further detailed analysis of each phosphorylation site of Rec8 and Mis4/Pds5 might be required for drawing firm conclusion."

Minor points:

- A sentence from the introduction is contradicting one in the results: "Cohesion is established through exchanging the loader complexes with a key cohesin accessory factor, Pds5, which recruits Eso1, a cohesin acetylase (Vaur et al. 2012; Goto et al. 2017)." versus "Once Psm3-K105K106 acetylation takes place, Mis4 is replaced by Pds5 because Pds5 but not Mis4 binds acetylated Psm3 (Petela et al. 2021)."

Psm3-K105K106 acetylation takes place during DNA replication by the action of PCNA-bound Eso1 rather than Pds5-bound Eso1 (GL Moldovan et al. *Molecular Cell*, 2006). We changed the words in the second sentence. "Once Psm3-K105K106 acetylation takes place during DNA replication (Moldovan et al. 2006), Mis4 is readily replaced by Pds5 because Pds5 but not Mis4 binds acetylated Psm3 (Petela et al. 2021)."

- There is no reference to Fig. 3D in the text.

We are sorry for this mistake. These sentences: "We analyzed phosphomimetic mutants psm3-T182E or psm3-S1001D. Although neither mutation showed defects in meiotic chromosome segregation in *rec12*⁺ cells, *psm3-ED* (T182E and S1001D) caused equational segregation in >70% cells." belong to Fig 3D and we added it.

- "non-phosphorylatable" is not a proper word.

This word has also been used in previous numerous papers including (Kitajima et al. 2006; Ma et al. 2021; Yoshida et al. 2022). We think this word is acceptable.

Kitajima TS, Sakuno T, Ishiguro K, Iemura S, Natsume T, Kawashima SA, Watanabe Y. 2006.

Shugoshin collaborates with protein phosphatase 2A to protect cohesin. *Nature* 441: 46-52.

Ma W, Zhou JW, Chen J, Carr AM, Watanabe Y. 2021. Meikin synergizes with shugoshin to protect cohesin Rec8 during meiosis I. *Genes & Development* 35: 692-697.

Yoshida A, Phillips-Mason P, Tarallo V, Avril S, Koivisto C, Leone G, Diehl JA. 2022.

Non-phosphorylatable cyclin D1 mutant potentiates endometrial hyperplasia and drives carcinoma with Pten loss. *Oncogene* 41: 2187-2195.

- The first sentence of the introduction could lead readers to think that cohesion is mediated by cohesin only in fission yeast. I recommend changing it to: "Sister chromatid cohesion of chromosomes is carried out by a multi-subunit complex, called cohesin, which in fission yeast comprises two SMC (structural maintenance of chromosome) family proteins, Psm1 and Psm3, a kleisin Rad21 subunit and an accessory subunit Psc3.

We followed the suggestion.

- It should be mentioned that separase "cleaves Rad21 along the entire chromosome" only in yeast. **We followed the suggestion and described as "To initiate anaphase, the anaphase-promoting complex (APC/C) triggers the degradation of securin, an inhibitory chaperone for separase that cleaves Rad21 along the entire chromosome in yeast."**

- There are two typos on page 2: "Mis4/Psd5".

We corrected them.

- The expression "phosphorylation in the either or all 15S/T sites" needs improvement.

We accepted the suggestion and described as "phosphorylation at some or all 15S/T sites"

- We identified five polo-kinase consensus sequences in these domains (Fig. 2E).

Thanks. We corrected this.

- Bottom of Fig. 2C: Shouldn't the genotype include *rec8-2A* and the frequency of equational segregation lowered from 100% to >80% to match the data shown in 2D?

We followed this suggestion.

We hope that these changes are satisfactory and that the revised manuscript is now acceptable.

Sincerely,

Yoshinori Watanabe, PhD

February 22, 2024

RE: Life Science Alliance Manuscript #LSA-2023-02556R

Prof. Yoshinori Watanabe
Jiangnan University
Science Center for Future Foods
1800 Lihu Avenue
Wuxi, Jiangsu 214122
China

Dear Dr. Watanabe,

Thank you for submitting your revised manuscript entitled "Phosphorylation of Rec8 cohesin complexes regulates mono-orientation of kinetochores in meiosis I". We would be happy to publish your paper in Life Science Alliance pending final revisions necessary to meet our formatting guidelines.

- please be sure that the authorship listing and order is correct
- please upload your main and supplementary figures as single files
- please upload your Table in editable .doc or Excel format
- please add ORCID ID for the corresponding author -- you should have received instructions on how to do so
- please add the Twitter handle of your host institute/organization as well as your own or/and one of the authors in our system
- please note that the titles in the system and on the manuscript file must match
- please move your main, supplementary figure, and table legends to the main manuscript text after the references section
- there is a call-out for Fig. 4F in the manuscript text, and the corresponding figure has no F panel -- please correct
- please add callouts for Figures S1A-C; S2A-D; S3A, B, D, E to your main manuscript text

FIGURE CHECKS:

- please add sizes next to all blots

A. FINAL FILES:

B. MANUSCRIPT ORGANIZATION AND FORMATTING:

Sincerely,

Reviewer #3 (Comments to the Authors (Required)):

In their revised manuscript, Watanabe and co-workers have adequately addressed all the previously raised points. In my point of view, the manuscript is now truly fit for publication in LSA.

February 27, 2024

RE: Life Science Alliance Manuscript #LSA-2023-02556RR

Prof. Yoshinori Watanabe
Jiangnan University
Science Center for Future Foods
1800 Lihu Avenue
Wuxi, Jiangsu 214122
China

Dear Dr. Watanabe,

Thank you for submitting your Research Article entitled "Phosphorylation of Rec8 cohesin complexes regulates mono-orientation of kinetochores in meiosis I". It is a pleasure to let you know that your manuscript is now accepted for publication in Life Science Alliance. Congratulations on this interesting work.

DISTRIBUTION OF MATERIALS:

Again, congratulations on a very nice paper. I hope you found the review process to be constructive and are pleased with how the manuscript was handled editorially. We look forward to future exciting submissions from your lab.

Sincerely,
